# Optimistic Regret Minimization for Extensive-Form Games via Dilated Distance-Generating Functions[*]

**Gabriele Farina**
Computer Science Department
Carnegie Mellon University
gfarina@cs.cmu.edu

**Christian Kroer**
IEOR Department
Columbia University
christian.kroer@columbia.edu

**Tuomas Sandholm**
Computer Science Department, CMU
Strategic Machine, Inc.
Strategy Robot, Inc.
Optimized Markets, Inc.
sandholm@cs.cmu.edu

## Abstract

We study the performance of optimistic regret-minimization algorithms for both minimizing regret in, and computing Nash equilibria of, zero-sum extensive-form games. In order to apply these algorithms to extensive-form games, a distance-generating function is needed. We study the use of the dilated entropy and dilated Euclidean distance functions. For the dilated Euclidean distance function we prove the first explicit bounds on the strong-convexity parameter for general treeplexes. Furthermore, we show that the use of dilated distance-generating functions enable us to decompose the mirror descent algorithm, and its optimistic variant, into local mirror descent algorithms at each information set. This decomposition mirrors the structure of the counterfactual regret minimization framework, and enables important techniques in practice, such as distributed updates and pruning of cold parts of the game tree. Our algorithms provably converge at a rate of $T^{-1}$, which is superior to prior counterfactual regret minimization algorithms. We experimentally compare to the popular algorithm CFR+, which has a theoretical convergence rate of $T^{-0.5}$ in theory, but is known to often converge at a rate of $T^{-1}$, or better, in practice. We give an example matrix game where CFR+ experimentally converges at a relatively slow rate of $T^{-0.74}$, whereas our optimistic methods converge faster than $T^{-1}$. We go on to show that our fast rate also holds in the Kuhn poker game, which is an extensive-form game. For games with deeper game trees however, we find that CFR+ is still faster. Finally we show that when the goal is minimizing regret, rather than computing a Nash equilibrium, our optimistic methods can outperform CFR+, even in deep game trees.

## 1 Introduction

Extensive-form games (EFGs) are a broad class of games that can model sequential interaction, imperfect information, and stochastic outcomes. To operationalize them they must be accompanied by techniques for computing game-theoretic equilibria such as Nash equilibrium. A notable success story of this is poker: Bowling et al. [1] computed a near-optimal Nash equilibrium for heads-up

---

[*]The full version of this paper is available on arXiv.

limit Texas hold'em, while Brown and Sandholm [3] beat top human specialist professionals at the larger game of heads-up no-limit Texas hold'em. Solving extremely large EFGs relies on many methods for dealing with the scale of the problem: abstraction methods are sometimes used to create smaller games [16, 26, 20, 14, 6, 21], endgame solving is used to compute refined solutions to the end of the game in real time [9, 15, 27], and recently depth-limited subgame solving has been very successfully used in real time [28, 8, 5]. At the core of all these methods is a reliance on a fast algorithm for computing approximate Nash equilibria of the abstraction, endgame, and/or depth-limited subgame [28, 8, 5]. In practice the most popular method has been the CFR$^+$ algorithm [38, 35], which was used within all three two-player poker breakthroughs [1, 28, 3]. CFR$^+$ has been shown to converge to a Nash equilibrium at a rate of $T^{-0.5}$, but in practice it often performs much better, even outperforming faster methods that have a guaranteed rate of $T^{-1}$ [7, 24, 23, 4].

Recently, another class of optimization algorithms has been shown to have appealing theoretical properties. *Online convex optimization* (OCO) algorithms are online variants of first-order methods: at each timestep $t$ they receive some loss function $\ell^t$ (often a linear loss which is a gradient of some underlying loss function), and must then recommend a point from some convex set based on the series of past points and losses. While these algorithms are generally known to have a $T^{-0.5}$ rate of convergence when solving static problems, a recent series of papers showed that when two *optimistic* OCO algorithms are faced against each other, and they have some estimate of the next loss faced, a rate of $T^{-1}$ can be achieved [30, 31, 34]. In this paper we investigate the application of these algorithms to EFG solving, both in the regret-minimization setting, and for computing approximate Nash equilibria at the optimal rate of $O(T^{-1})$. The only prior attempt at using optimistic OCO algorithm in extensive-form games is due to Farina et al. [13]. In that paper, the authors show that by restricting to the weaker notion of *stable-predictive optimism*, one can mix and match local stable-predictive optimistic algorithm at every decision point in the game as desired and obtain an overall stable-predictive optimistic algorithm that enables $O(T^{-0.75})$ convergence to Nash equilibrium. The approach we adopt in this paper is different from that of Farina et al. [13] in that our construction does not allow one to pick different regret minimizers for different decision points; however, our algorithms converge to Nash equilibrium at the improved rate $O(T^{-1})$.

The main hurdle to overcome is that in all known OCO algorithms a *distance-generating function* (DGF) is needed to maintain feasibility via proximal operators and ensure that the stepsizes of the algorithms are appropriate for the convex set at hand. For the case of EFGs, the convex set is known as a *treeplex*, and the so-called dilated DGFs are known to have appealing properties, including closed-form iterate updates and strong convexity properties [18, 24]. In particular, the dilated entropy DGF, which applies the negative entropy at each information set, is known to lead to the state-of-the-art theoretical rate on convergence for iterative methods [24]. Another potential DGF is the dilated Euclidean DGF, which applies the $\ell_2$ norm as a DGF at each information set. We show the first explicit bounds on the strong-convexity parameter for the dilated Euclidean DGF when applied to the strategy space of an EFG. We go on to show that when a dilated DGF is paired with the *online mirror descent* (OMD) algorithm, or its optimistic variant, the resulting algorithm decomposes into a recursive application of local online mirror descent algorithms at each information set of the game. This decomposition is similar to the decomposition achieved in the counterfactual regret minimization framework, where a local regret minimizer is applied on the counterfactual regret at each information set. This localization of the updates along the tree structure enables further techniques, such as distributing the updates [3, 6] or skipping updates on cold parts of the game tree [2].

It is well-known that the entropy DGF is the theoretically superior DGF when applied to optimization over a simplex [18]. For the treeplex case where the entropy DGF is used at each information set, Kroer et al. [24] showed that the strong theoretical properties of the simplex entropy DGF generalize to the dilated entropy DGF on a treeplex (with earlier weaker results shown by Kroer et al. [22]). Our results on the dilated Euclidean DGF confirm this finding, as the dilated Euclidean DGF has a similar strong convexity parameter, but with respect to the $\ell_2$ norm, rather than the $\ell_1$ norm for dilated entropy (having strong convexity with respect to the $\ell_1$ norm leads to a tighter convergence-rate bound because it gives a smaller matrix norm, another important constant in the rate).

In contrast to these theoretical results, for the case of computing a Nash equilibrium in matrix games it has been found experimentally that the Euclidean DGF often performs much better than the entropy DGF. This was shown by Chambolle and Pock [11] when using a particular accelerated primal-dual algorithm [10, 11] and using the *last iterate* (as opposed to the uniformly-averaged iterate as the

theory suggests). Kroer [19] recently showed that this extends to the theoretically-sound case of using linear or quadratic averaging in the same primal-dual algorithm, or in mirror prox [29] (the offline variant of optimistic OMD). In this paper we replicate these results when using OCO algorithms: first we show it on a particular matrix game, where we also exhibit a slow $T^{-0.74}$ convergence rate of CFR$^+$ (the slowest CFR$^+$ rate seen to the best of our knowledge). We show that for the Kuhn poker game the last iterate of optimistic OCO algorithms with the dilated Euclidean DGF also converges extremely fast. In contrast to this, we show that for deeper EFGs CFR$^+$ is still faster. Finally we compare the performance of CFR$^+$ and optimistic OCO algorithms for minimizing regret, where we find that OCO algorithms perform better.

## 2 Regret Minimization Algorithms

In this section we present the regret-minimization algorithms that we will work with. We will operate within the framework of *online convex optimization* [37]. In this setting, a decision maker repeatedly plays against an unknown environment by making decision $\boldsymbol{x}^1, \boldsymbol{x}^2, \ldots \in \mathcal{X}$ for some convex compact set $\mathcal{X}$. After each decision $\boldsymbol{x}^t$ at time $t$, the decision maker faces a *linear loss* $\boldsymbol{x}^t \mapsto \langle \boldsymbol{\ell}^t, \boldsymbol{x}^t \rangle$, where $\boldsymbol{\ell}^t$ is a vector in $\mathcal{X}$. Summarizing, the decision maker makes a decision $\boldsymbol{x}^{t+1}$ based on the sequence of losses $\boldsymbol{\ell}^1, \ldots, \boldsymbol{\ell}^t$ as well as the sequence of past iterates $\boldsymbol{x}^1, \ldots, \boldsymbol{x}^t$.

The quality metric for a regret minimizer is its *cumulative regret*, which is the difference between the loss cumulated by the sequence of decisions $\boldsymbol{x}^1, \ldots, \boldsymbol{x}^T$ and the loss that would have been cumulated by playing the best-in-hindsight time-independent decision $\hat{\boldsymbol{x}}$. Formally, the cumulative regret up to time $T$ is

$$R^T := \sum_{t=1}^{T} \langle \boldsymbol{\ell}^t, \boldsymbol{x}^t \rangle - \min_{\hat{\boldsymbol{x}} \in \mathcal{X}} \left\{ \sum_{t=1}^{T} \langle \boldsymbol{\ell}^t, \hat{\boldsymbol{x}} \rangle \right\}.$$

A "good" regret minimizer is such that the cumulative regret grows *sublinearly in $T$*.

The algorithms we consider assume access to a *distance-generating function* $d : \mathcal{X} \to \mathbb{R}$, which is 1-strongly convex (with respect to some norm) and continuously differentiable on the interior of $\mathcal{X}$. Furthermore $d$ should be such that the gradient of the convex conjugate $\nabla d(\boldsymbol{g}) = \operatorname{argmax}_{\boldsymbol{x} \in \mathcal{X}} \langle \boldsymbol{g}, \boldsymbol{x} \rangle - d(\boldsymbol{x})$ is easy to compute. Following Hoda et al. [18] we say that a DGF satisfying these properties is a *nice* DGF for $\mathcal{X}$. From $d$ we also construct the *Bregman divergence* $D(\boldsymbol{x} \parallel \boldsymbol{x}') := d(\boldsymbol{x}) - d(\boldsymbol{x}') - \langle \nabla d(\boldsymbol{x}'), \boldsymbol{x} - \boldsymbol{x}' \rangle$.

First we present two classical regret minimization algorithms. The *online mirror descent* (OMD) algorithm produces iterates according to the rule

$$\boldsymbol{x}^{t+1} = \operatorname*{argmin}_{\boldsymbol{x} \in \mathcal{X}} \left\{ \langle \boldsymbol{\ell}^t, \boldsymbol{x} \rangle + \frac{1}{\eta} D(\boldsymbol{x} \parallel \boldsymbol{x}^t) \right\}. \tag{1}$$

The *follow the regularized leader* (FTRL) algorithm produces iterates according to the rule [32]

$$\boldsymbol{x}^{t+1} = \operatorname*{argmin}_{\boldsymbol{x} \in \mathcal{X}} \left\{ \left\langle \sum_{\tau=1}^{t} \boldsymbol{\ell}^{\tau}, \boldsymbol{x} \right\rangle + \frac{1}{\eta} d(\boldsymbol{x}) \right\}. \tag{2}$$

OMD and FTRL satisfy regret bounds of the form $R^T \leq O\left( D(\boldsymbol{x}^* \| \boldsymbol{x}^1) L \sqrt{T} \right)$ (e.g. Hazan [17]).

The *optimistic* variants of the classical regret minimization algorithms take as input an additional vector $\boldsymbol{m}^{t+1}$, which is an estimate of the loss faced at time $t+1$ [12, 30]. Optimistic OMD produces iterates according to the rule [30] (note that $\boldsymbol{x}^{t+1}$ is produced before seeing $\boldsymbol{\ell}^{t+1}$, while $\boldsymbol{z}^{t+1}$ is produced after)

$$\boldsymbol{x}^{t+1} = \operatorname*{argmin}_{\boldsymbol{x} \in \mathcal{X}} \left\{ \langle \boldsymbol{m}^{t+1}, \boldsymbol{x} \rangle + \frac{1}{\eta} D(\boldsymbol{x} \parallel \boldsymbol{z}^t) \right\}, \quad \boldsymbol{z}^{t+1} = \operatorname*{argmin}_{\boldsymbol{z} \in \mathcal{X}} \left\{ \langle \boldsymbol{\ell}^{t+1}, \boldsymbol{z} \rangle + \frac{1}{\eta} D(\boldsymbol{z} \parallel \boldsymbol{z}^t) \right\}. \tag{3}$$

Thus it is like OMD, except that $\boldsymbol{x}^{t+1}$ is generated by an additional step taken using the loss estimate. This additional step is transient in the sense that $\boldsymbol{x}^{t+1}$ is not used as a center for the next iterate.

OFTRL produces iterates according to the rule [30, 34]

$$\boldsymbol{x}^{t+1} = \operatorname*{argmin}_{\boldsymbol{x} \in \mathcal{X}} \left\{ \left\langle \boldsymbol{m}^{t+1} + \sum_{\tau=1}^{t} \boldsymbol{\ell}^{\tau}, \boldsymbol{x} \right\rangle + \frac{1}{\eta} d(\boldsymbol{x}) \right\}. \tag{4}$$

Again the loss estimate is used in a transient way: it is used as if we already saw the loss at time $t+1$, but then discarded and not used in future iterations.

## 2.1 Connection to Saddle Points

A *bilinear saddle-point problem* is a problem of the form $\min_{\boldsymbol{x} \in \mathcal{X}} \max_{\boldsymbol{y} \in \mathcal{Y}} \{\boldsymbol{x}^{\top} \boldsymbol{A} \boldsymbol{y}\}$, where $\mathcal{X}, \mathcal{Y}$ are closed convex sets. This general formulation allows us to capture, among other settings, several game-theoretical applications such as computing Nash equilibria in two-player zero-sum games. In that setting, $\mathcal{X}$ and $\mathcal{Y}$ are convex polytopes whose description is provided by the *sequence-form constraints*, and $\boldsymbol{A}$ is a real payoff matrix [36].

The error metric that we use is the *saddle-point residual* (or *gap*) $\xi$ of $(\bar{\boldsymbol{x}}, \bar{\boldsymbol{y}})$, defined as $\xi(\bar{\boldsymbol{x}}, \bar{\boldsymbol{y}}) := \max_{\hat{y} \in \mathcal{Y}} \langle \bar{\boldsymbol{x}}, \boldsymbol{A} \hat{\boldsymbol{y}} \rangle - \min_{\hat{x} \in \mathcal{X}} \langle \hat{\boldsymbol{x}}, \boldsymbol{A} \bar{\boldsymbol{y}} \rangle$. A well-known folk theorem shows that the average of a sequence of regret-minimizing strategies for the choice of losses $\boldsymbol{\ell}_{\mathcal{X}}^{t} : \mathcal{X} \ni \boldsymbol{x} \mapsto (-\boldsymbol{A}\boldsymbol{y}^{t})^{\top} x$, $\boldsymbol{\ell}_{\mathcal{Y}}^{t} : \mathcal{Y} \ni \boldsymbol{y} \mapsto (\boldsymbol{A}^{\top} \boldsymbol{x}^{t})^{\top} \boldsymbol{y}$ leads to a bounded saddle-point residual, since one has

$$\xi(\bar{\boldsymbol{x}}, \bar{\boldsymbol{y}}) = \frac{1}{T}(R_{\mathcal{X}}^{T} + R_{\mathcal{Y}}^{T}). \tag{5}$$

When $\mathcal{X}, \mathcal{Y}$ are the players' sequence-form strategy spaces, this implies that the average strategy profile produced by the regret minimizers is a $1/T(R_{\mathcal{X}}^{T} + R_{\mathcal{Y}}^{T})$-Nash equilibrium. This also implies that by using online mirror descent or follow-the-regularizer-leader, one obtains an anytime algorithm for computing a Nash equilibrium. In particular, at each time $T$, the average strategy output by each of the two regret minimizers forms a $\epsilon$-Nash equilibrium, where $\epsilon = O(T^{-0.5})$.

## 2.2 RVU Property and Fast Convergence to Saddle Points

Both optimistic OMD and optimistic FTRL satisfy the *Regret bounded by Variation in Utilities* (RVU) property, as given by Syrgkanis et al.:

**Definition 1** (RVU property, [34]). *We say that a regret minimizer satisfies the RVU property if there exist constants $\alpha > 0$ and $0 < \beta \le \gamma$, as well as a pair of dual norms $(\|\cdot\|, \|\cdot\|_*)$ such that, no matter what the loss functions $\boldsymbol{\ell}^1, \ldots, \boldsymbol{\ell}^T$ are,*

$$R^{T} \le \alpha + \beta \sum_{t=1}^{T} \|\boldsymbol{\ell}^{t} - \boldsymbol{m}^{t}\|_{*}^{2} - \gamma \sum_{t=1}^{T} \|\boldsymbol{x}^{t} - \boldsymbol{x}^{t-1}\|^{2}. \tag{RVU}$$

The definition given here is slightly more general than that of Syrgkanis et al. [34]: we allow a general estimate $\boldsymbol{m}^t$ of $\boldsymbol{\ell}^t$, whereas their definition requires using $\boldsymbol{m}^t = \boldsymbol{\ell}^{t-1}$. While the choice $\boldsymbol{m}^t = \boldsymbol{\ell}^{t-1}$ is often reasonable, in some cases other definitions of the loss prediction are more natural [13]. In practice, both optimistic OMD and optimistic FTRL satisfy a parametric notion of the RVU property, which depends on the value of the step-size parameter that was chosen to set up either algorithm.

**Theorem 1** (Syrgkanis et al. [34]). *For all step-size parameters $\eta > 0$, Optimistic OMD satisfies the RVU conditions with respect to the primal-dual norm pair $(\|\cdot\|_1, \|\cdot\|_\infty)$ with parameters $\alpha = R/\eta, \beta = \eta, \gamma = 1/(8\eta)$, where $R$ is a constant that scales with the maximum allowed norm of any loss function $\ell$.*

**Theorem 2.** *For all step-size parameters $\eta > 0$, OFTRL satisfies the RVU conditions with respect to any primal-dual norm pair $(\|\cdot\|, \|\cdot\|_*)$ with parameters $\alpha = \Delta_d/\eta, \beta = \eta, \gamma = 1/(4\eta)$, where $\Delta_d := \max_{\boldsymbol{x}, \boldsymbol{y} \in \mathcal{X}} \{d(\boldsymbol{x}) - d(\boldsymbol{y})\}$.*

Our proof, available in the appendix of the full paper, generalizes the work by Syrgkanis et al. [34] by extending the proof beyond simplex domains and beyond the fixed choice $\boldsymbol{m}^t = \boldsymbol{\ell}^{t-1}$.

It turns out that this is enough to accelerate the convergence to a saddle point in the construction of Section 2.1. In particular, by letting the predictions be defined as $\boldsymbol{m}_{\mathcal{X}}^t := \boldsymbol{\ell}_{\mathcal{X}}^{t-1}, \boldsymbol{m}_{\mathcal{Y}}^t := \boldsymbol{\ell}_{\mathcal{Y}}^{t-1}$, we

obtain that the residual $\xi$ of the average decisions $(\bar{\boldsymbol{x}}, \bar{\boldsymbol{y}})$ satisfies

$$
T\xi(\bar{\boldsymbol{x}}, \bar{\boldsymbol{y}}) \leq \frac{2\alpha'}{\eta} + \eta \sum_{t=1}^{T} \left( \| -\boldsymbol{A}\boldsymbol{y}^t + \boldsymbol{A}\boldsymbol{y}^{t-1} \|_*^2 + \| \boldsymbol{A}^\top \boldsymbol{x}^t - \boldsymbol{A}^\top \boldsymbol{x}^{t-1} \|_*^2 \right)
$$
$$
- \frac{\gamma'}{\eta} \sum_{t=1}^{T} \left( \| \boldsymbol{x}^t - \boldsymbol{x}^{t-1} \|^2 + \| \boldsymbol{y}^t - \boldsymbol{y}^{t-1} \|^2 \right)
$$
$$
\leq \frac{2\alpha'}{\eta} + \left( \eta \| \boldsymbol{A} \|_{\mathrm{op}}^2 - \frac{\gamma'}{\eta} \right) \left( \sum_{t=1}^{T} \| \boldsymbol{x}^t - \boldsymbol{x}^{t-1} \|^2 + \sum_{t=1}^{T} \| \boldsymbol{y}^t - \boldsymbol{y}^{t-1} \|^2 \right),
$$

where the first inequality holds by plugging (RVU) into (5), and the second inequality by noting that the operator norm $\| \cdot \|_{\mathrm{op}}$ of a linear function is equal to the operator norm of its transpose. This implies that when the step-size parameter is chosen as $\eta = \frac{\sqrt{\gamma'}}{\| \boldsymbol{A} \|_{\mathrm{op}}}$, the saddle-point gap $\xi(\bar{\boldsymbol{x}}, \bar{\boldsymbol{y}})$ satisfies $\xi(\bar{\boldsymbol{x}}, \bar{\boldsymbol{y}}) \leq \frac{2\alpha' \| \boldsymbol{A} \|_{\mathrm{op}}}{T \sqrt{\gamma'}} = O(T^{-1})$.

## 3 Treeplexes and Sequence Form

We formalize a sequential decision process as follows. We assume that we have a set of decision points $\mathcal{J}$. Each decision point $j \in \mathcal{J}$ has a set of actions $A_j$ of size $n_j$. Given a specific action at $j$, the set of possible decision points that the agent may next face is denoted by $\mathcal{C}_{j,a}$. It can be an empty set if no more actions are taken after $j, a$. We assume that the decision points form a tree, that is, $\mathcal{C}_{j,a} \cap \mathcal{C}_{j',a'} = \emptyset$ for all other convex sets and action choices $j', a'$. This condition is equivalent to the perfect-recall assumption in extensive-form games, and to conditioning on the full sequence of actions and observations in a finite-horizon partially-observable decision process. In our definition, the decision space starts with a root decision point, whereas in practice multiple root decision points may be needed, for example in order to model different starting hands in card games. Multiple root decision points can be modeled by having a dummy root decision point with only a single action.

The set of possible next decision points after choosing action $a \in A_j$ at decision point $j \in \mathcal{J}$, denoted $\mathcal{C}_{j,a}$, can be thought of as representing the different decision points that an agent may face after taking action $a$ and then making an observation on which she can condition her next action choice. In addition to games, our model of sequential decision process captures, for example, partially-observable Markov decision processes and Markov decision processes where we condition on the entire history of observations and actions.

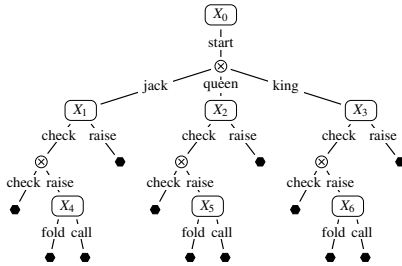

Figure 1: Sequential action space for the first player in the game of Kuhn poker. $\otimes$ denotes an observation point; $\bullet$ represents the end of the decision process.

As an illustration, consider the game of Kuhn poker [25]. Kuhn poker consists of a three-card deck: king, queen, and jack. The action space for the first player is shown in Figure 1. For instance, we have: $\mathcal{J} = \{0, 1, 2, 3, 4, 5, 6\}$; $n_0 = 1$; $n_j = 2$ for all $j \in \mathcal{J} \setminus \{0\}$; $A_0 = \{\text{start}\}$, $A_1 = A_2 = A_3 = \{\text{check}, \text{raise}\}$, $A_4 = A_5 = A_6 = \{\text{fold}, \text{call}\}$; $\mathcal{C}_{0,\text{start}} = \{1, 2, 3\}$, $\mathcal{C}_{1,\text{raise}} = \emptyset$, $\mathcal{C}_{3,\text{check}} = \{6\}$; etc.

The expected loss for a given strategy is non-linear in the vectors of probability masses for each decision point $j$. This non-linearity is due to the probability of reaching each $j$, which is computed as the product of the probabilities of all actions on the path to from the root to $j$. An alternative formulation which preserves linearity is called the *sequence form*. In the sequence-form representation, the simplex strategy space at a generic decision point $j \in \mathcal{J}$ is scaled by the decision variable associated with the last action in the path from the root of the process to $j$. In this formulation, the value of a particular action represents the probability of playing the whole *sequence* of actions from the root to that action. This allows each term in the expected loss to be weighted only by the sequence ending in the corresponding action. The sequence form has been used to instantiate linear programming [36] and first-order methods [18, 22, 24] for computing Nash equilibria of zero-sum EFGs. Formally, the sequence-form representation $\mathcal{X}$ of a sequential decision process can

be obtained recursively, as follows: for every $j \in \mathcal{J}, a \in A_j$, we let $\mathcal{X}_{\downarrow j,a} := \prod_{j' \in \mathcal{C}_{j,a}} \mathcal{X}_{\downarrow j'}$, where $\Pi$ denotes Cartesian product; at every decision point $j \in \mathcal{J}$, we let

$$\mathcal{X}_{\downarrow j} := \{(\lambda_1, \ldots, \lambda_{n_j}, \lambda_1 \boldsymbol{x}_{a_1}, \ldots, \lambda_{n_j} \boldsymbol{x}_{a_{n_j}}) : (\lambda_1, \ldots, \lambda_n) \in \Delta^{n_j}, \boldsymbol{x}_a \in \mathcal{X}_{\downarrow j,a} \ \forall a \in A_j\},$$

where we assumed $A_j = \{a_1, \ldots, a_{n_j}\}$.

The sequence form strategy space for the whole sequential decision process is then $\mathcal{X} := \{1\} \times \mathcal{X}_{\downarrow r}$, where $r$ is the root of the process. The first entry, identically equal to 1 for any point in $\mathcal{X}$, corresponds to what is called the *empty sequence*. Crucially, $\mathcal{X}$ is a convex and compact set, and the expected loss of the process is a linear function over $\mathcal{X}$. With the sequence-form representation the problem of computing a Nash equilibrium in an EFG can be formulated as a *bilinear saddle-point problem* (see Section 2.1), where $\mathcal{X}$ and $\mathcal{Y}$ are the sequence-form strategy spaces of the sequential decision processes faced by the two players, and $\boldsymbol{A}$ is a sparse matrix encoding the leaf payoffs of the game.

As we have already observed, vectors that pertain to the sequence form have one entry for each sequence of the decision process. We denote with $v_\phi$ the entry in $\boldsymbol{v}$ corresponding to the empty sequence, and $v_{ja}$ the entry corresponding to any other sequence $(j, a)$ where $j \in \mathcal{J}, a \in A_j$. Sometimes, we will need to *slice* a vector $\boldsymbol{v}$ and isolate only those entries that refer to all decision points $j'$ and actions $a' \in A_{j'}$ that are at or below some $j \in \mathcal{J}$; we will denote such operation as $\boldsymbol{v}_{\downarrow j}$. Similarly, we introduce the syntax $v_j$ to denote the subset of $n_j = |A_j|$ entries of $\boldsymbol{v}$ that pertain to all actions $a \in A_j$ at decision point $j \in \mathcal{J}$. Finally, note that for any $j \in \mathcal{J} - \{r\}$ there is a unique sequence $(j', a')$, denoted $p_j$ and called *the parent sequence of decision point $j$*, such that $j \in \mathcal{C}_{j'a'}$. When $j = r$ is the root decision point, we let $p_r := \phi$, the empty sequence.

## 4 Dilated Distance Generating Functions

We will be interested in a particular type of DGF which is suitable for sequential decision-making problems: a *dilated DGF*. A dilated DGF is constructed by taking a sum over suitable local DGFs for each decision point, where each local DGF is dilated by the parent variable leading to the decision point: $d(\boldsymbol{x}) = \sum_{j \in \mathcal{J}} x_{p_j} d_j\left(\frac{\boldsymbol{x}_j}{x_{p_j}}\right)$. Each "local" DGF $d_j$ is given the local variable $\boldsymbol{x}_j$ divided by $x_{p_j}$, so that $\frac{\boldsymbol{x}_j}{x_{p_j}} \in \Delta^{n_j}$. The idea is that $d_j$ can be any DGF suitable for $\Delta^{n_j}$; by multiplying $d_j$ by $x_{p_j}$ and taking a sum over $\mathcal{J}$ we construct a DGF for the whole treeplex from these local DGFs. Hoda et al. [18] showed that dilated DGFs have many of the desired properties of a DGF for an optimization problem over a treeplex.

We now present two local DGFs for simplexes, that are by far the most common in practice. In the following we let $\boldsymbol{b}$ be a vector in the $n$-dimensional simplex $\Delta^n$. First, the *Euclidean DGF* $d(\boldsymbol{b}) = \|\boldsymbol{b}\|_2^2$, which is 1-strongly convex with respect to the $\ell_2$ norm; secondly, the *negative entropy DGF* $d(\boldsymbol{b}) = \sum_{i=1}^n b_i \log(b_i)$ (we will henceforth drop the "negative" and simply refer to it as the entropy DGF), which is 1-strongly convex with respect to the $\ell_1$ norm. The strong convexity properties of the dilated entropy DGF were shown by Kroer et al. [24] (with earlier weaker results shown by Kroer et al. [22]). However, for the dilated Euclidean DGF a setup for achieving a strong-convexity parameter of 1 was unknown until now; Hoda et al. [18] show that a strong-convexity parameter exists, but do not show what it is for the general case (they give specific results for a particular class of *uniform treeplexes*). We now show how to achieve this.

We are now ready to state our first result on dilated regularizers that are strongly convex with respect to the Euclidean norm:

**Theorem 3.** *Let $d(\boldsymbol{x}) = \sum_{j \in \mathcal{J}} x_{p_j} d_j(\boldsymbol{x}_j/x_{p_j})$ where for all $j$, $d_j$ is $\mu_j$-strongly convex with respect to the Euclidean norm over $\Delta^{n_j}$. Furthermore, define $\sigma_{ja} := \frac{\mu_j}{2} - \sum_{j' \in C_{ja}} \mu_{j'}$, and $\bar{\sigma} := \min_{ja} \sigma_{ja}$. Then, $d$ is $\bar{\sigma}$-strongly convex with respect to the Euclidean norm over $\mathcal{X}$.*

We can immediately use Theorem 3 to prove the following corollary:

**Corollary 1.** *Let $\bar{\sigma} > 0$ be arbitrary, and for all $j$ let $d_j$ be a $\mu_j$-strongly convex function over $\Delta^{n_j}$ with respect to the Euclidean norm, where the $\mu_j$'s satisfy*

$$\mu_j = 2\bar{\sigma} + 2 \max_{a \in A_j} \sum_{j' \in C_{ja}} \mu_{j'}. \tag{6}$$

*Then, $d(\boldsymbol{x}) = \sum_{j \in \mathcal{J}} x_{p_j} d_j(\boldsymbol{x}_j/x_{p_j})$ is $\bar{\sigma}$-strongly convex over $\mathcal{X}$ with respect to the Euclidean norm.*

# 5 Local Regret Minimization

We now show that OMD and Optimistic OMD run on a treeplex $\mathcal{X}$ with a dilated DGF can both be interpreted as locally minimizing a modified variant of loss at each information set, with correspondingly-modified loss predictions. The modified local loss at a given information set $j$ takes into account the loss and DGF below $j$ by adding the expectation with respect to the next iterate $\boldsymbol{x}_{\downarrow j}^t$. In practice this modified loss is easily handled by computing $\boldsymbol{x}^t$ bottom-up, thereby visiting $j$ after having visited the whole subtree below.

We first show that the problem of computing the *prox mapping*, the minimizer of a linear term plus the Bregman divergence, decomposes into local prox mappings at each simplex of a treeplex. This will then be used to show that OMD and Optimistic OMD can be viewed as a tree of local simplex-instantiations of the respective algorithms.

## 5.1 Decomposition into Local Prox Mappings with a Dilated DGF

We will be interested in solving the following prox mapping, which takes place in the sequence form:

$$\text{Prox}(\boldsymbol{g}, \hat{\boldsymbol{x}}) = \operatorname*{argmin}_{\boldsymbol{x} \in \mathcal{X}} \left\{ \langle \boldsymbol{g}, \boldsymbol{x} \rangle + D(\boldsymbol{x} \parallel \hat{\boldsymbol{x}}) \right\}. \tag{7}$$

The reason is that the update applied at each iteration of several OCO algorithms run on the sequence-form polytope of $\mathcal{X}$ can be described as an instantiation of this prox mapping. We now show that this update can be interpreted as a local prox mapping at each decision point, but with a new loss $\hat{g}_j$ that depends on the update applied in the subtree beneath $j$.

**Proposition 1** (Decomposition into local prox mappings). *A prox mapping* (7) *on a treeplex with a Bregman divergence constructed from a dilated DGF decomposes into local prox mappings at each decision point $j$ where the solution is as follows:*

$$\boldsymbol{x}_j^* = x_{p_j} \cdot \operatorname*{argmin}_{\boldsymbol{b}_j \in \Delta^{n_j}} \left\{ \langle \hat{\boldsymbol{g}}_j, \boldsymbol{b}_j \rangle + D_j \left( \boldsymbol{b}_j \; \middle\| \; \frac{\hat{\boldsymbol{x}}_j}{\hat{x}_{p_j}} \right) \right\},$$

*where*

$$\hat{g}_{j,a} = g_{j,a} + \sum_{j' \in \mathcal{C}_{j,a}} \left[ d_{\downarrow j'}^* \left( -\boldsymbol{g}_{\downarrow j'} + \nabla d_{\downarrow j'}(\hat{\boldsymbol{x}}_{\downarrow j'}) \right) - d_{j'} \left( \frac{\hat{\boldsymbol{x}}_j}{\hat{x}_{p_j}} \right) + \left\langle \nabla d_{j'} \left( \frac{\hat{\boldsymbol{x}}_{j'}}{\hat{x}_{p_{j'}}} \right), \frac{\hat{\boldsymbol{x}}_{j'}}{\hat{x}_{p_{j'}}} \right\rangle \right].$$

Hoda et al. [18] and Kroer et al. [23] gave variations on a similar result: that the convex conjugate $d_{\downarrow j}^*(-\boldsymbol{g})$ can be computed in bottom-up fashion similar to the recursion we show here. Proposition 1 is slightly different in that we additionally show that the Bregman divergence also survives the decomposition and can be viewed as a local Bregman divergence. This latter difference will be necessary for showing that OMD can be interpreted as a local RM.

## 5.2 Decomposition into Local Regret Minimizers

With Proposition 1 it follows almost directly that OMD and Optimistic OMD can be seen as a set of local regret minimizers, one for each simplex. Each produces iterates from their respective simplex, with the overall strategy produced by then applying the sequence-form transformation to these local iterates.

**Theorem 4.** *OMD with a dilated DGF for a treeplex $\mathcal{X}$ corresponds to running OMD locally at each simplex $j$, with the local loss $\hat{\boldsymbol{\ell}}^t$ constructed according to Proposition 1. Optimistic OMD corresponds to the optimistic variant of this local OMD with local loss predictions $\hat{\boldsymbol{\ell}}^t, \hat{\boldsymbol{m}}_j^{t+1}$ again constructed according to Proposition 1 using $\boldsymbol{x}^t$ as Bregman divergence center and $\boldsymbol{x}^{t+1}$ for aggregating losses below each simplex. Here the modified loss uses $\boldsymbol{z}_{\downarrow j'}^t$ and $\boldsymbol{x}^{t+1}$ as Bregman divergence center and aggregating loss below, respectively. The prediction $\hat{\boldsymbol{m}}_j^{t+1}$ uses $\boldsymbol{z}_{\downarrow j'}^t$ and $\boldsymbol{z}^{t+1}$.*

Unlike OMD and its optimistic variant, it is not the case that FTRL has a nice interpretation as a local regret minimizer. The reason is that the prox mapping in (2) or (4) minimizes the sum of losses, rather than the most recent loss. Because of this, the expected value $\langle \sum_{\tau=1}^t \boldsymbol{\ell}_{\downarrow j}^\tau, \boldsymbol{x}_{\downarrow j}^{t+1} \rangle$ at simplex $j$, which

influences the modified loss at parent simplexes, is computed based on $x^{t+1}$ for all $t$ losses. Thus there is no local modified loss that could be received at rounds $1$ through $t$ that accurately reflects the modified loss needed in Proposition 1.

## 6  Experimental Evaluation

We experimentally evaluate the performance of optimistic regret minimization methods instantiated with dilated distance-generating functions. We experiment on three games:

- *Smallmatrix*, a small $2 \times 2$ matrix game. Given a mixed strategy $x = (x_1, x_2) \in \Delta^2$ for Player 1 and a mixed strategy $y = (y_1, y_2) \in \Delta^2$ for Player 2, the payoff function for player 1 is $u(x, y) = 5x_1y_1 - x_1y_2 + x_2y_2$.

- Kuhn poker, already introduced in Section 3. In Kuhn poker, each player first has to put a payment of $1$ into the pot. Each player is then dealt one of the three cards, and the third is put aside unseen. A single round of betting then occurs: first, Player 1 can check or bet $1$. Then,
    - If Player 1 checks Player 2 can check or raise $1$.
        * If Player 2 checks a showdown occurs; if Player 2 raises Player 1 can fold or call.
            · If Player 1 folds Player 2 takes the pot; if Player 1 calls a showdown occurs.
    - If Player 1 raises Player 2 can fold or call.
        * If Player 2 folds Player 1 takes the pot; if Player 2 calls a showdown occurs.
    If no player has folded, a showdown occurs where the player with the higher card wins.

- Leduc poker, a standard benchmark in imperfect-information game solving [33]. The game is played with a deck consisting of 5 unique cards with 2 copies of each, and consists of two rounds. In the first round, each player places an ante of $1$ in the pot and receives a single private card. A round of betting then takes place with a two-bet maximum, with Player 1 going first. A public shared card is then dealt face up and another round of betting takes place. Again, Player 1 goes first, and there is a two-bet maximum. If one of the players has a pair with the public card, that player wins. Otherwise, the player with the higher card wins. All bets in the first round are $1$, while all bets in the second round are $2$. This game has 390 decision points and 911 sequences per player.

**Fast Last-Iterate Convergence.** In the first set of experiments (Figure 2, top row), we compare the saddle-point gap of the strategy profiles produced by optimistic OMD and optimistic FTRL to that produced by CFR and CFR$^+$. Optimistic OMD and optimistic FTRL were set up with the step-size parameter $\eta = 0.1$ in Smallmatrix and $\eta = 2$ in Kuhn Poker, and the plots show the last-iterate convergence for the optimistic algorithms, which has recently received attention in the works by Chambolle and Pock [11] and Kroer [19]. Finally, we instantiated optimistic OMD and optimistic FTRL with the Euclidean distance generating function as constructed in Corollary 1. The plots show that—at least in these shallow games—optimistic methods are able to produce even up to 12 orders of magnitude better-approximate saddle-points than CFR and CFR$^+$.

Interestingly, Smallmatrix appears to be a hard instance for CFR$^+$: linear regression on the first $20\,000$ iterations of CFR$^+$ shows, with a coefficient of determination of roughly $0.96$, that $\log \xi(\boldsymbol{x}_*^T, \boldsymbol{y}_*^T) \approx -0.7375 \cdot \log(T) - 2.1349$, where $(\boldsymbol{x}_*^T, \boldsymbol{y}_*^T)$ is the average strategy profile (computed using linear averaging, as per $CFR^+$'s construction) up to time $T$. In other words, we have evidence of at least one game in which the approximate saddle-point computed by CFR$^+$ experimentally has residual bounded below by $\Omega(T^{-0.74})$. This observation suggests that the analysis of CFR$^+$ might actually be quite tight, and that CFR$^+$ is *not* an accelerated method.

Figure 2 (bottom left) shows the performance of OFTRL in Leduc Poker, compared to CFR and CFR$^+$ (we do not show optimistic OMD, which we found to have worse performance than OFTRL). Here OFTRL performs worse than CFR$^+$. This shows that in deeper games, more work has to be done to fully exploit the accelerated bounds of optimistic regret minimization methods.

**Comparing the Cumulative Regret.** We also compared the algorithms based on the sum of cumulative regrets (again we omit optimistic OMD, which performed worse than OFTRL). In all three games, OFTRL leads to lower sum of cumulative regrets. Figure 2 (bottom right) shows the performance of OFTRL in Leduc Poker. Here, we used the usual average of iterates $\bar{x} := 1/T \sum_{t=1}^{T} x^t$ (note that the choice of averaging strategy has no effect on the bottom right plot.)

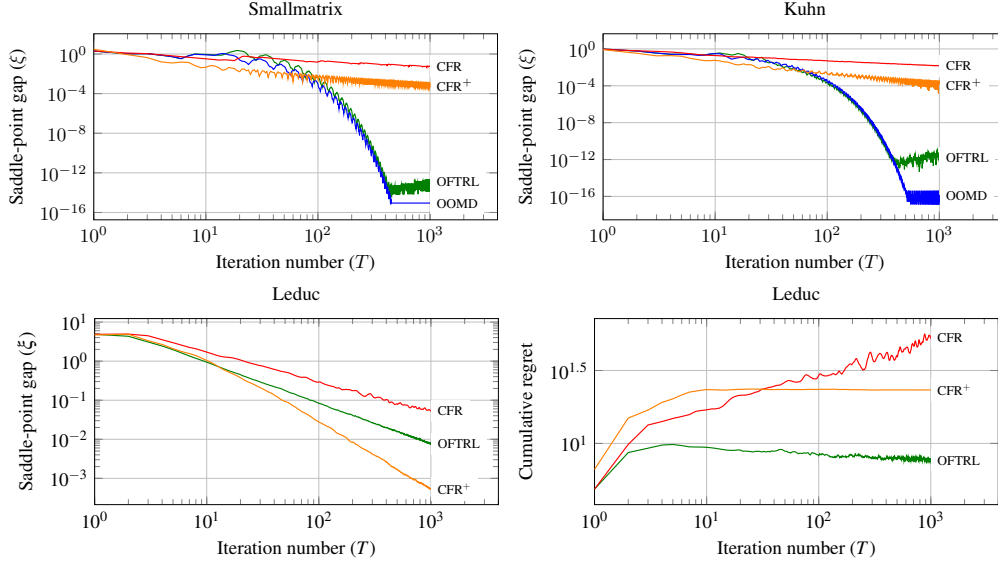

Figure 2: (Left and upper right) Saddle-point gap as a function of the number of iterations. The plots show the last-iterate convergence for OOMD and OFTRL.(Lower right) Sum of cumulative regret for both players in Leduc. Optimistic OMD (OOMD) and OFTRL use step-size parameter $\eta = 0.1$ in Smallmatrix and $\eta = 2$ in Kuhn. OFTRL uses step-size parameter $\eta = 200$ in Leduc.

OFTRL's performance matches the theory from Theorem 2 and Section 2.2. In particular, we observe that while OFTRL does not beat the state-of-the-art CFR$^+$ in terms of saddle-point gap, it beats it according to the regret sum metric. The fact that CFR$^+$ performs worse with respect to the regret sum metric is somewhat surprising: the entire derivation of CFR and CFR$^+$ is based on showing bounds on the regret sum. However, the connection between regret and saddle-point gap (or exploitability) is one-way: if the two regret minimizers (one per player) have regret $R_1$ and $R_2$, then the saddle point gap can be easily shown to be less than or equal to $(R_1 + R_2)/T$. However, nothing prevents it from being much smaller than $(R_1 + R_2)/T$. What we empirically find is that for CFR$^+$ this bound is very loose. We are not sure why this is the case, and it potentially warrants further investigation in the future.

# 7  Conclusions

We studied how optimistic regret minimization can be applied in the context of extensive-form games, and introduced the first instantiations of regret-based techniques that achieve $T^{-1}$ convergence to Nash equilibrium in extensive-form games. These methods rely crucially on having a tractable regularizer to maintain feasibility and control the stepsizes on the domain at hand—in our case, the sequence-form polytope. We provided the first explicit bound on the strong convexity properties of dilated distance-generating functions with respect to the Euclidean norm. We also showed that when optimistic regret minimization methods are instantiated with dilated distance-generating functions, the regret updates are local to each information set in the game, mirroring the structure of the counterfactual regret minimization framework. This localization of the updates along the tree structure enables further techniques, such as distributing the updates or skipping updates on cold parts of the game tree. Finally, when used in self play, these optimistic regret minimization methods guarantee an optimal $T^{-1}$ convergence rate to Nash equilibrium.

We demonstrate that in shallow games, methods based on optimistic regret minimization can significantly outperform CFR and CFR$^+$—even up to 12 orders of magnitude. In deeper games, more work has to be done to fully exploit the accelerated bounds of optimistic regret minimization methods. However, while the strong CFR$^+$ performance in large games remains a mystery, we elucidate some points about its performance—including showing that its theoretically slow convergence bound is somewhat tight. Finally, we showed that when the goal is minimizing regret, rather than computing a Nash equilibrium, optimistic methods can outperform CFR$^+$ even in deep game trees.

## Acknowledgments

This material is based on work supported by the National Science Foundation under grants IIS-1718457, IIS-1617590, and CCF-1733556, and the ARO under award W911NF-17-1-0082. Gabriele Farina is supported by a Facebook fellowship.

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
