[Reviews · NeurIPS 2019]

Reviewer 1



Edit after author response <<< No concerns after respons and proposed changes for camera ready. I would still like to see OOMD results in addition to the discussion proposed in the response. >>> Given the amount of theory, the paper was fairly clearly written. It extends the recent optimistic regret minimization framework to sequential decision making strategies (treeplexes) -- extending the state of the art for a moderately popular theoretical framework. The paper uses the optimistic regret framework with two saddlepoint solving algorithms, and compares the game solving algorithms to a currently used algorithm CFR+, showing at least one game instance suggesting CFR+ is asymptotically worse in the worst case.

Reviewer 2



I think the method is interesting, since the use of dilated distance functions it allows interpreting the Mirror Descent type iteration as one of minimizing local regret (at the cost of growing the state space). In addition, a theoretical result of strong convexity is obtained in the case of the Euclidean norm. As far as I can see, the mathematical arguments seem correct. However, I have two questions, regarding the results: -I do not understand where is used the flexibility given by the choice of the $m^t$. This is important, since part of the theoretical work consists in adapting some known results to that case. What kind of vector did you use in these cases? Without further use, it seems that this have been done because it was technically feasible. -The fact that OOMD-type methods do not work in deep games (but those of the OFTRL type do) implies that decomposition as local minimization does not seem to have an important experimental effect. So what could be the advantage of such idea in the general case? %%%%% After Rebuttal After reading the authors' responses and looking at the others reviewers comments and discussions, I agree to change my score. This is a good submission.

Reviewer 3



------------ Edit after reading the other reviews and the rebuttal: Thank you for answering my questions. I have no concerns. ------------ I have a strong background with CFR / CFR+, but not any in accelerated methods, optimistic regret matching, or the other theoretical components of this work. Through that lens, I liked this paper. This paper considers the application of optimistic regret-minimization algorithms to extensive form games. It follows a similar setup as CFR, the current SOTA algorithm for solving large games, which decomposes the overall regret-minimization problem into independent subproblems at each information set. Theoretically, the new algorithms should have faster convergence than the CFR+ variant of CFR. Empirically, the authors show that these methods converge dramatically faster than CFR+ in small games, but slower in the slightly larger game of Leduc hold'em. I found that the paper was clearly motivated and described. To my knowledge it is novel, and significant: CFR / CFR+ has had a long run as the SOTA algorithm for solving large extensive games, and any technique (such as this one) with stronger theoretical bounds and faster convergence (even if only in smaller games) is of interest. The paper is well structured with ample discussion of each component. While I'm sure that I didn't appreciate all of the details without a theoretical background in this area, I think I could probably implement it from this description. Overall I don't have any substantive issues with this work. I've listed some questions in the Improvements section.

[Author Response · NeurIPS 2019]

**Response to all**  We thank the reviewers for their nice and thorough reviews. All reviewers mention variants of the following question: **"No optimistic OMD results for the Leduc Poker domain"**. We omitted OOMD from Figure 2 (bottom row) because we were not able to trigger the interesting property that OFTRL enjoys: simultaneously having a worse saddle-point gap and a better cumulative regret, as compared to CFR+. In fact, we found OOMD's performance in Leduc3 to be significantly worse than OFTRL's, both in terms of saddle-point gap and regret. Understanding these qualitative differences between the two algorithms is an interesting direction of research. We will include this discussion in the final version of the paper.

**Response to Reviewer #1**

- **"Section 3, line 187"** Yes, it was a typo. Thanks!
- **"Leduc Poker experiments shown in the bottom row, the OOMD and OFTL results are still for the last iterate?"** No, we used the "standard" average $\bar{x} := \frac{1}{T}\sum_{t=1}^{T} x^t$ of all iterates $x^1, \ldots, x^T$, in line with the theory (Line 153). Note that the choice of averaging strategy has no effect on the bottom right plot. We will include this discussion in the final version of the paper.
- **"Comparing the Cumulative Regret (and Conclusions)"** Even if our paper mainly uses regret minimizers as a way to compute a saddle point, our regret minimizers can be used in other contexts too. For example, any of our regret minimizers can be used as an online decision maker that plays against an opponent controlled by the environment (see for example (Farina et al., 2019a) cited in our paper, where they use this approach to find exploitative strategies in games). In those settings, regret is the meaningful quality metric of the regret minimizer.
- Fair point regarding both places where we say "prove" but really mean "give relatively conclusive experimental evidence." We will update both.

**Response to Reviewer #2**

- **Re "...where is used the flexibility given by the choice of the $m^t$."** There are at least two cases in which being able to freely choose $m^t$ (rather than setting it to the fixed choice $m^t = \ell^{t-1}$) helps:
  - Beyond saddle-point solving: even if our paper uses regret minimizers as a way to compute a saddle point, these regret minimizers can be used in other contexts too. For example, any of our regret minimizers can be used as an online decision maker that plays against an opponent controlled by the environment; in that case, if a statistical model of the opponent is available, the best prediction of the next loss might very well be different from $\ell^{t-1}$.
  - In other saddle-point algorithms: for example, Farina et al.[1] show how to combine different optimistic regret minimizers and obtain a composite regret minimizer that they use to find a saddle point in two-player zero-sum extensive-form games. They found that being able to pick $m^t$ to something other than $\ell^{t-1}$ is beneficial in their construction (see top left column on page 7, as well as Equation 17 in their paper).
- **Re "...OOMD-type methods do not work in deep games (but those of the OFTRL type do)..."** OOMD and OFTRL both work in deep games, in the sense that they are guaranteed to converge to a saddle point at a rate of $O(T^{-1})$. But experimentally we did find worse performance for OOMD; see the response to all.
- **Re "...decomposition...advantage of such idea in the general case?"** We think the decomposition is interesting for at least two reasons. First, it is the first accelerated method that allows a "CFR-like" interpretation of the method, in the sense that updates are local. From a theoretical perspective we think this is interesting in its own right. Second, it enables a lot of practical experimentation. While this would not technically retain the theoretical rate, it would be interesting to find ways to incorporate ideas such as stepsizes that adapt at a local level, pruning, and other "local" ideas that have been popular in CFR variants.

**Response to Reviewer #3**

- **Re "... only OFTRL in Leduc?"** See response to all.
- **"not the Entropy DGF used in the empirical results?"** See lines 71-83 in our paper for the reason why we are more interested in the Euclidean DGF than entropy. We have experiments with the entropy DGF as well. We can add them to the appendix for future versions of the paper. They are consistent with the observations for other settings that we mention in lines 71-83; it is worse than Euclidean DGF.
- **Re "Is there no hope that a similar effect would occur in Leduc if run longer?"** It's possible but we think it's unlikely. If that were to be the case, we think it would happen at such high precision that it would not be interesting from a practical perspective.
- **Re "... relationship between minimizing regret and exploitability ..."** The connection between regret and saddle-point gap (or exploitability) is one-way: if the two regret minimizers (one per player) have regret $R_1$ and $R_2$, then the saddle point gap can be easily shown to be $\leq R_1 + R_2$. However, nothing prevents it from being *much* smaller than $R_1 + R_2$. What we empirically found is that for CFR$^+$ this bound is very loose. We are not sure as to why this is the case, and we agree that it is an interesting fact that deserves more investigation in the future.

## Footnotes

[1]Farina, Kroer, Brown and Sandholm. Stable-Predictive Optimistic Counterfactual Regret Minimization. AAAI 2019.


[Meta-Review · NeurIPS 2019]

This paper proposes the use of "dilated" distance generating functions to write the associated mirror descent algorithm as a local regret minimization procedure on the decision nodes in two-person zero-sum extensive form games. After the discussion phase, the reviewers and myself did not have any major concerns for the paper and I am happy to recommend acceptance (modulo the reviewers' specific suggestions for points to be revised in the camera-ready).